# On the use of Cortical Magnification and Saccades as Biological Proxies for Data Augmentation

**Binxu Wang**[1], **David Mayo**[3],[\*] **Arturo Deza**[4], **Andrei Barbu**[5], **Colin Conwell**[2]
[1]Dept. of Neurobiology, Harvard Medical School &
Dept. of Neuroscience, Washington University in St Louis
[2]Dept. of Psychology, Harvard University; [3]Google Research, Brain Team
[4]BCS & CBMM, MIT; [5]CSAIL & CBMM, MIT

## Abstract

Self-supervised learning is a powerful way to learn useful representations from natural data. It has also been suggested as one possible means of building visual representation in humans, but the specific objective and algorithm are unknown. Currently, most self-supervised methods encourage the system to learn an invariant representation of different transformations of the same image in contrast to those of other images. However, such transformations are generally non-biologically plausible, and often consist of contrived perceptual schemes such as random cropping and color jittering. In this paper, we attempt to reverse-engineer these augmentations to be more biologically or perceptually plausible while still conferring the same benefits for encouraging robust representation. Critically, we find that random cropping can be substituted by cortical magnification, and saccade-like sampling of the image could also assist the representation learning. The feasibility of these transformations suggests a potential way that biological visual systems could implement self-supervision. Further, they break the widely accepted spatially-uniform processing assumption used in many computer vision algorithms, suggesting a role for spatially-adaptive computation in humans and machines alike. Our code and demo can be found here (Wang, 2021).

## 1 Introduction

It has long been suggested that the human visual system is tuned not solely via supervised learning, but also with the assistance of innate inductive biases and more recently self-supervised learning schemes. If so, how might we approximate the power of self-supervised learning in biological vision? A natural place to begin is with the recent progress in self-supervised learning in machine vision.

Recent years have seen great advances in self-supervised learning systems (Chen et al., 2020a; Grill et al., 2020) and their application to machine (Geirhos et al., 2020) and human vision (Konkle & Alvarez, 2020; Orhan et al., 2020b) alike. One common theme in these methods are the learning pressures that push the learned representations to be invariant to certain augmentations. These transformations are manually picked, and their effects differ substantially in producing a robust visual representation(Chen et al., 2020a). Random crop and color jittering have been shown empirically to be good transformations in the sense that using them within a contrastive framework yields both high in-distribution generalization and robustness to common corruptions (Hendrycks et al., 2019; Naseer et al., 2020).

---

[\*]Work completed as part of the Google AI Residency Program

3rd Workshop on Shared Visual Representations in Human and Machine Intelligence (SVRHM 2021) of the Neural Information Processing Systems (NeurIPS) conference, Virtual.

The triumph of the general principle of learning invariant representation and the success of the specific transforms of random crop and color jittering suggest that biological visual system may have also evolved to leverage the same objective for self-supervised learning. In other words, to generate invariant representations against augmentations may be one goal our visual system uses to train itself in order to learn object recognition. If so, what could be the augmentations the visual system learns to be invariant against? Two biological processes appear to be relevant: 1) the spatially-varying resolution (foveation) at the retina (Anstis, 1974); and 2) the active sampling procedure *(saccades)* which happen naturally three times per second (Eckstein, 2011; Traver & Bernardino, 2010). The combination of the two processes creates a frequently changing retinal input to our visual system, while our perception of the world remains stable. It's a reasonable objective to encourage the high level visual representation to remain stable across views of the environment. Indeed, there is neurobiological evidence that higher visual cortices have longer time constants than lower visual cortex, suggesting a more stable representation of natural input (Chaudhuri et al., 2015). Thus we hypothesize that these foveation-based transformations and saccade-like active sampling mechanisms may be the natural augmentations that biological visual systems learn to be invariant against, forming a counterpart to the random crops in the self-supervised learning framework for computer vision.

Indeed, recent work has suggested a functional goal of the adaptive multi-resolution nature of the primate retina – mainly tailored towards achieving scale invariance (Poggio et al., 2014; Han et al., 2020), adversarial robustness (Luo et al., 2015; Reddy et al., 2020; Jonnalagadda et al., 2021) and o.o.d. generalization (Deza & Konkle, 2020). Given these hints of a representational goal of foveation – beyond computational efficiency – we hypothesize that it could possibly give rise to similar or better performance in terms of generalization than its non-biologically plausible counterparts, random cropping and color jittering, within a self-supervised framework.

In this work, we work to test these natural augmentations computationally, i.e. to jointly implement foveation and saccades as augmentations under a self-supervised learning framework, and then compare it against their computer vision counter-part augmentations (e.g. random crops). Altogether, this work provides a proof of concepts for whether it's possible to leverage these natural augmentations in the service of learning robust visual representations without semantic labels.

## 2 Methods

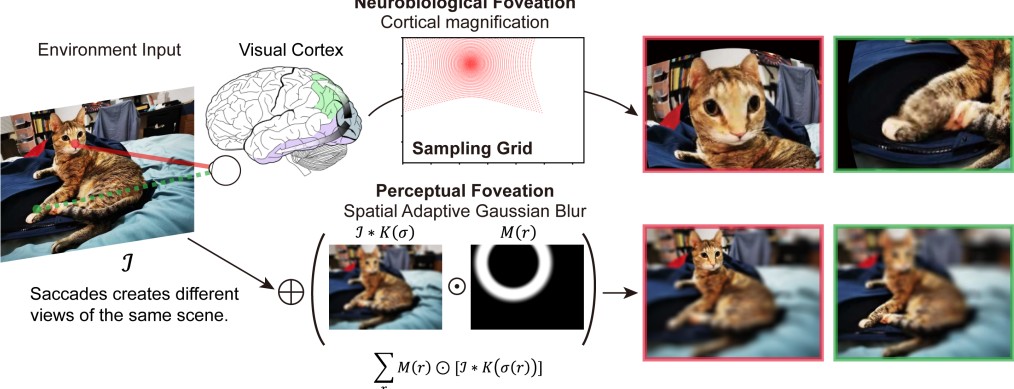

Figure 1: **Foveation Transform**. The upper row shows foveation as magnification, which models the cortical representation of the foveated image. We implemented this operation by sampling from a warped grid around the fixation point. The lower row shows foveation as adaptive blur, which models the perceptual experience of the foveated imagae. It's implemented by blending images blurred by different kernels with different masks.

Our natural augmentation pipeline consists of a saccade (selection of fixation point) and a foveation transform (emphasizing visual information around the fixation).

We implement saccades as sampling multiple **fixation points** on the same image. These fixation points $p = (x, y)$ are generated by i.i.d. sampling either from a uniform distribution over the center part of the image, or from a gaze probability distribution over the image. Conditional probability between fixation points are not considered in this work, but worth development in the future. In our implementation, we pre-computed the saliency maps $S$ of all images with the FastSal model (Hu

& McGuinness, 2021) pre-trained on the SALICON dataset (Jiang et al., 2015). This saliency map is interpreted as the unnormalized log-density of the gaze probability $P[x, y]$ on the image. We continuously tune the effect of the saliency information by changing a "temperature" parameter $T$ of the sampling density.

$$P[x, y] \propto \exp(\frac{S[x, y] - \max(S)}{T})$$

Higher temperature will tune down the effect of saliency and approximate a uniform distribution over the image, closer to the sampler of the original random crop. Lower temperature will amplify the effect of saliency, resulting in a focused, low-entropy sampling of highly salient parts of the object (Fig.2E). Note that given the specification of FastSal model, $T = 1$ shall give back the fixation density distribution that is similar to the training dataset.

We now move to foveation, which we define as spatially-adaptive (non-uniform) processing of the image (Eckstein, 2011; Bajcsy et al., 2018). We identified at least two approaches to implement foveation as a transform of the images, one inspired by our perceptual experience of foveation, the other inspired by the structure of the visual cortical map (Fig.1). We compared both of them in our experiments.

**Foveation as Spatial-Varying Blur** This approach is inspired by our visual perception of the world: when we fixate our gaze, our perception of the visual periphery seems lower resolution than the central field of view. Moreover, the whole scene seems stable regardless of the shift of the fixation point. To approximate this phenomena, people have used spatial varying blur (Geisler & Perry, 1998; Pramod et al., 2018; Malkin et al., 2020) or texture-based distortions (Freeman & Simoncelli, 2011; Rosenholtz et al., 2012; Deza et al., 2017; Wallis et al., 2019; Kaplanyan et al., 2019) to transform an image such that it matches our perception of it. The extent of distortion or Gaussian blur changes as a function of the eccentricity.

We implemented this foveation transform $\mathcal{F}$ as classically done in Geisler & Perry (1998), and also Pramod et al. (2018); Malkin et al. (2020): Given a fixation point, the pixels in the image are divided into belts of similar eccentricity, represented by masks $M_i$ (Deza & Eckstein, 2016); and the image is convolved with Gaussian kernels of different standard deviation $\sigma_i$, corresponding to the degree of blur at different eccentricity; finally, these blurred images $I * K(\sigma_i)$ are blended into one with the belt-shape masks (Fig.1).

$$\mathcal{F}(I) = \sum_i^{N_b} M_i \cdot [I * K(\sigma_i)], \quad \sum_i^{N_b} M_i[i, j] = 1$$

This was implemented using simple operations in computer vision (convolution, pixel-wise multiplication and addition) such that it could be an efficient step in the augmentation pipeline during training. The standard deviation of the blur kernel is a linear function of the eccentricity $\sigma_i = K e_i$; additionally, the pixels within eccentricity $e_o$ are not blurred, corresponding to foveal vision. So, there are two parameters: the size of the foveal area (ratio of unblurred pixels) $fov\ area$, and the linear coefficient $K$ of kernel size with respect to eccentricity. For more details, see Sec.A.1.

**Foveation as Cortical Magnification** This approach is inspired by the structure of retinotopic maps on the visual cortex esp. primary visual cortex: due to the difference density of retina ganglion cells in central and peripheral vision(Wässle et al., 1989), the cortical area corresponding to the unit area of retinal image varies as a function of eccentricity(Van Essen et al., 1984). Thus, we can think of the image in cortical coordinates as a warped version of the corresponding retinal image (Fig.1). When our eyes move around on a scene, the cortical "image" will shift with our gaze Yates et al. (2021) and magnify different parts depending on the fixation points; and the viewing distance will affect the magnification factor. If we assume the typical CNN as a model of cortical visual processing, then the cortical magnified views of an image is a reasonable transformation to use in representation learning (Bashivan et al., 2019).

Classically, the cortical magnification factor is an inverse function of the eccentricity, (Harvey & Dumoulin, 2011). Additionally, here we assume the CMF is constant close to fovea since the data there is usually lacking.

$$CMF(r) = \begin{cases} C & r < r_{fov} \\ \frac{C(r+r_{fov})}{r+K} & r \geq r_{fov} \end{cases}$$

Here, we transform the retinal eccentricity to cortical radial distance by integrating the reciprocal of the cortical magnification function. Thus, we get the retinal eccentricity $e(r)$ as a piecewise linear (foveal) or quadratic (peripheral) function of the cortical radius $r$. For further analysis of this function and the fitting of human cortical retinotopy data, see Sec. A.2 and Fig. A.1.

$$e(r) = \int_0^r \frac{1}{CMF(\rho)} d\rho = \frac{1}{C} \begin{cases} r; & r < r_{fov} \\ \frac{(r+K)^2}{2(r_{fov}+K)} + \frac{r_{fov}-K}{2}; & r \geq r_{fov} \end{cases}$$

We assume an isotropic transform from retinal images to cortical images, so we can use this radial transform to build the 2d change of coordinates $(r, \theta) \mapsto (e(r), \theta)$. We note that the constant $1/C$ controls the scaling of the map, which affects the image area covered by the sampling grid. Thus we use the **cover ratio** as the control parameter in experiments. Besides, parameter $r_{fov}$ controls the area of linear sampling while $K$ controls the degree of distortion in the periphery, which are manually tuned and fixed ($r_{fov} = 30, K = 20$).

## 3 Experiments and Results

We adapted the framework of Sim-CLRv2 (Chen et al., 2020b), and used the STL-10 dataset (96 pixel resolution) as our test bed to evaluate the effectiveness of the proposed transforms. ResNet18 models (He et al., 2016) were trained with the default settings of SimCLR (see A.4) on the unlabeled set (100K images) of the STL-10 dataset. We evaluated the quality of representation by linear probes: For every 5 epochs, we froze the model and trained a linear classifier from its representation vector to the labels with the training set (500 images× 10 classes), and then evaluated this linear classifier on the testing set (800 images× 10 classes). The training and test set accuracy and the evolution of each across the training procedure are reported as our major evaluation criteria. (Note that on this dataset, linear evaluation of a random visual representation (randomly initialized ResNet18) can already achieve around 50% training accuracy and 39% test accuracy.) We also take note of another statistic, top-1 SimCLR accuracy, the accuracy of the self-supervised learning task.

| crop | fov | blur | fov area | train | test | SimCLR |
|---|---|---|---|---|---|---|
| ✓* | | ✓ | | 86.6 | 80.5 | 78.1 |
| ✓† | | ✓ | | 86.5 | 80.8 | 74.3 |
| ✓ | | ✓ | | 86.6 | 80.7 | 79.2 |
| ✓ | ✓ | | [0.01, 0.1] | 84.4 | 78.2 | 78.6 |
| ✓ | ✓ | | [0.01, 0.5] | 84.1 | 78.3 | 78.9 |
| ✓ | ✓ | | [0.1, 0.5] | 83.6 | 77.8 | 79.0 |
| | ✓ | | [0.01, 0.1] | 40.4 | 38.4 | 100.0 |
| | ✓ | | [0.01, 0.5] | 39.6 | 36.8 | 100.0 |
| | ✓ | | [0.1, 0.5] | 41.0 | 39.0 | 100.0 |
| | | ✓ | | 38.0 | 35.0 | 99.7 |

Table 1: **Performance of Foveation as Blur Augmentation**. **Crop** denotes the random resized crop augmentation guided by pre-computed saliency maps; with the exceptions: ∗ used the original crop, † used flat saliency map. **Fov** refers to foveation as blur transform. Train, test accuracy at 96 epochs and the exponential moving averaged ($\alpha = 0.6$) SimCLR objective accuracy at 99 epochs are reported, same below.

| crop | mag | blur | cover ratio | train | test | SimCLR |
|---|---|---|---|---|---|---|
| ✓ | | ✓ | | 85.2 | 79.7 | 84.8 |
| | ✓ | | [0.01, 0.35] | 84.4 | 79.6 | 59.4 |
| | ✓ | | [0.05, 0.35] | 85.5 | 79.7 | 74.8 |
| | ✓ | | [0.05, 0.7] | 81.7 | 76.4 | 86.2 |
| | ✓ | | [0.01, 1.5] | 78.6 | 72.6 | 89.6 |

Table 2: **Performance of Foveation as Magnification Augmentation**. Here **crop** denotes the original random resized crop. **Magn** denotes the foveation as magnification transform. Statistics of row 1 and 3 are averaged across 12 runs while the rest are averaged across 2 runs.

This is the success rate of distinguishing views of the same image from those of different images; in our setting when batch size is 256 this is the accuracy of a 511-way classification. We interpret this statistic as reflecting the intrinsic difficulty of the augmentation: the lower the SimCLR accuracy, the more challenging the augmentation is.

In the following experiments, we kept the common augmentations in place: random horizontal flip, random color jittering, and randomly turning images into gray scale. With blur, we refer to the spatial uniform Gaussian blur of the image.

**Experiment 1: Foveation as Adaptive Blur** First, we tested if the foveation as adaptive blur could substitute random crops as a natural alternative. In this experiment, fixation points were uniformly

sampled. We found that, using either foveation or uniform blur without random crop, the SimCLR training failed: the final test accuracy (36.8-39.0%) is close to or even worse than that of random networks (39.0%) (Tab. 1). Using foveation in addition to random crop will result in similar and or worse representation quality (77.8-78.3%) as the baseline using blur (80.6%). Note that without random crop, SimCLR accuracy is saturated. We found that after a few epochs this saturated objective no longer guides the learning of representation ; in other words, with variations only in color and resolution but not space, learning the equivalence of different views of same image is trivial for a CNN which is built to learn local image features. In this regard, "moving" the image with respect to the field of view of the CNN seems necessary towards learning a good representation.

**Experiment 2: Foveation as Magnification** Next, we tested if foveation as magnification could replace random crop. We found that the magnification transform could indeed substitute the effects of random crop, with all other augmentations in place. We performed rough parameter tuning for the cortical magnification transform, and picked the best performing parameter ($fov = 30$, $K = 20$). We replicated the magnification transform and the random crop baseline $N = 12$ times and compared their representation quality. The test set accuracy for magnification transforms is $79.65 \pm 0.37\%$ (mean±std) while that for the random crop group is $79.72 \pm 0.23\%$. The 95% confidence interval of their difference is [-0.33, 0.17]%, which is not a significant difference in terms of overall accuracy, suggesting proper magnification is a biologically-plausible alternative to the random crop. Further, we conducted systematic parameter sweep and examined how the shape parameters affect the representation quality in Sec.A.3. We found that a larger $K$ and $fov$ value which created a less warped image, generally improved the learned representation. Besides, the cover ratio of range $[0.05, 0.35]$ consistently outperformed the other three ranges, for all the $K, fov$ values (Tab.2,A.1). We reasoned that the image patches of this size do not overlap unreasonably, so the SimCLR task remains challenging and instructive; these images patches are also not too small, so they are still informative of the object category (see discussion in Sec.A.3).

**Experiment 3: Random-dom Crops Guided by Saliency Maps** After finding the biological plausible equivalent of random crops, we moved on to test the effect of different models of saccade, i.e. the sampling distribution of the center of views. Specifically, we manipulated the temperature of the sampling, which controls the entropy of the fixation distribution. We found that with lower temperature the accuracy for SimCLR objective can also saturate (97.4% for 0.01 temperature or 93.9% for 0.1 temperature, Fig.2A) However, the linear evaluated accuracy either on training set or test set decreased as the temperature became too low (Fig.2B). In the other extreme, high enough

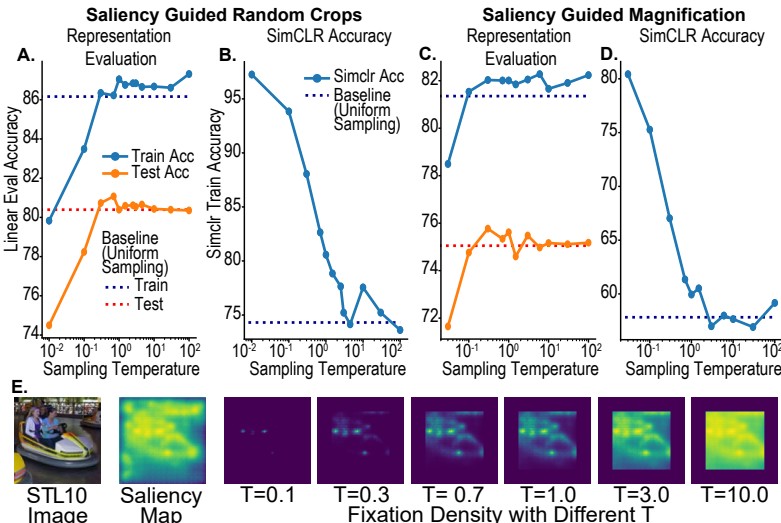

Figure 2: **Effect of sampling temperature on training and feature quality**. For **random crops** with different temperature (Exp 3), **A.** Linear evaluation of the representation quality as a function of temperature, uniform sampling baselines are shown in dashed lines; **B.** Final accuracy for SimCLR objective as a function of temperature. In the same format, **C.D.** show the effect of temperature, for **foveation as magnification** (Exp 4). **E.** Sample from STL-10, its saliency map and the fixation densities corresponding to different temperatures.

temperature will eliminate the effect of saliency map, and bring the test accuracy and SimCLR accuracy closer to the uniform sampling baseline. In between, there was a sweet spot at which the saliency guided sampling could improve the quality of learned representation: when temperature was in the range $[0.3, 4.5]$, the test accuracy increased about 0.3%-0.7%.

**Experiment 4: Foveation as Magnification Guided by Saliency Maps** Similar to experiment 3, we also performed saliency guided sampling for the magnification transform and examined the effect of sampling temperature. The results showed a similar trend as that in experiment 3 (Fig.2 C,D): sampling temperature has a U-shaped effect on the quality of learnt representation, with the optimal temperature falling in the range $[0.3, 1.5]$.

## 4 Discussion

Our results showed that foveation as spatially varying blur of the image *per se* is not a particularly strong mode of data augmentation, and its effect is comparable to uniform Gaussian blurring of the image. Reframing random crop as cortical magnification, however, provides both a biologically plausible method of data augmentation and similarly robust benefits to training. Furthermore, random sampling of the fixation points (via crops or cortical magnification; Exp 3, 4) based on pre-trained saliency information has a small but reasonable effect on the learned representation. Overall, finding these pattern of results are encouraging despite the lack of temporal dynamics (Akbas & Eckstein, 2017), inhibition-of-return or actual human fixation data (Koehler et al., 2014). Broadly, we think this computational framework should set the stage to further incorporate these and other factors in additional analyses where perceptual dimensions such as adversarial robustness and o.o.d. generalization can be tested.

The difference between foveation as blur and foveation as cortical magnification is intriguing. For self-supervised learning, it showed the importance of learning similar representations for spatially varying views (which was lacking in the foveated blur). In the parameter sweep of magnification transform (Tab. A.1), we further showed the importance of sampling local views with small overlaps. For visual neuroscience, this offers a potential mechanism for visual stability across saccades: if the visual system learns its representations in such a self-supervised fashion, then the learning objective could facilitate a stable representation or even an entire percept of the scene.

Altogether, this work presents another step on the journey to parity between classically used transformations in computer vision and biologically motivated computations that occur in human visual cortex. As previously discussed, future work should focus on incorporating the temporal nature of visual perception within a contrastive framework and comparing these learned representations with real human fixation data. Such directions have been growing in popularity as modern datasets have been tracking infant visual behavior (Sullivan et al., 2020; Orhan et al., 2020a). We think these datasets and approaches may provide an excellent first step for creating more 'infant'-like self-supervised learning regimes to gradually close the gap between human and machine perception. Finally, our results suggest a symbiotic motivation to continue to find biologically-plausible transformations for modern computer vision training pipelines rather than relying on hand-engineered heuristics.

## Acknowledgments and Disclosure of Funding

We appreciate the intellectual support and resources from Brain, Minds and Machines summer school. We are grateful for the inspiring discussions with the lecturers and colleagues during the summer school: Haim Sompolinsky, Thomas Serre, Mengmi Zhang, Zhiwei Ding. Thanks for Yunyi Shen from UWMadison for helpful discussion on the mathematical formulations of cortical magnification. We are thankful to the RIS cluster and staff in Washington University in St Louis for their generous provision of GPU resources and technical support.

B.W. is funded by the McDonnell Center for Systems Neuroscience in Washington University (pre-doctoral fellowship to B.W.). A.D. was funded by MIT's Center for Brains, Minds and Machines and Lockheed Martin Corporation. We declared no competing interests with this work.

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

# A Appendix

## A.1 Implementation Details for Foveation as Blur Transform

The equations to generate the belts around fovea are borrowed from Equ. 1,3 in Deza & Eckstein (2016). It reads

$$f(x) = \begin{cases} \cos^2(\pi(x + \frac{1}{4})); & \text{if } -\frac{3}{4} \le x \le -\frac{1}{4} \\ 1; & \text{if } -\frac{1}{4} \le x \le \frac{1}{4} \\ 1 - \cos^2(\pi(x - \frac{3}{4})); & \text{if } \frac{1}{4} \le x \le \frac{3}{4} \\ 0; & \text{otherwise} \end{cases}$$

$$g_n(e) = f(\frac{\log(e) - [\log(e_0) + w_e(n+1)]}{w_e}); \; w_e = \frac{\log(e_r) - \log(e_0)}{N_e}$$

The belt shaped masks are generated by passing the eccentricity (distance to fixation point $(x, y)$) of each pixel into $g_n(e)$ function

$$M_n[i, j] = g_n(\sqrt{(i - x)^2 + (j - y)^2})$$

## A.2 Rationale and Physiological Data for Cortical Magnification

To support the proposed cortical magnification, we reviewed retinotopy data for V1 in human subjects (Engel et al., 1997; Qiu et al., 2006; Wu et al., 2012). In these studies, the researchers measured the cortical responses to different sizes of ring gratings with fMRI to estimate the retinotopy of V1. We fetched the subject averaged data points (from Figure 9 of (Qiu et al., 2006)) and fit the retinal eccentricity-retinal distance relationship with the classic exponential model and our linear-quadratic model (Eq. 2).

We noted that the classic exponential fit would not cross the x axis, i.e. no cortical point would correspond to the exact foveal point. This is clearly an artifact of the exponential functional form, and a result of lacking data for activations to small images (< 2 deg). This made the classic exponential form unsuitable for our radial transform, thus we decided to derive our own radial transform function. Given the functional form in Eq. 2, with $C, K, r_{fov}$ as free parameter, we found we could fit the data points from (Qiu et al., 2006) well (Fig.A.1): the $R^2$ was 0.9999, with parameter $K = -7.73$ with 95% confidence interval $[-10.0, -5.44]$, $r_{fov} = 15.2$ with 95% confidence interval $[14.0, 16.4]$. Since $C$ is the overall scaling and unit dependent so we neglect it. In comparison the classic exponential fit had an $R^2$ of 0.9989. This analysis showed that our proposed function form for cortical magnification is expressive enough to accommodate existing human cortical magnification data.

Finally, note that the Eq. 2 could be nondimensionalized as

$$e(\tilde{r}) = \frac{1}{\tilde{C}} \begin{cases} \tilde{r}; & \tilde{r} < 1 \\ \frac{(\tilde{r} + \tilde{K})^2}{2(1 + \tilde{K})} + \frac{1 - \tilde{K}}{2}; & \tilde{r} \ge 1 \end{cases} \tag{1}$$

$$\tilde{r} = r/r_{fov}, \; \tilde{K} = K/r_{fov}, \; \tilde{C} = C/r_{fov} \tag{2}$$

Since the value of $r_{fov}$ and $C$ are influenced by the units of retinal eccentricity or cortical distance, $\tilde{K}$ is the unitless parameter that could be compared between the model and the human data. In Fig. A.1, $\tilde{K}$ is around -0.509.

## A.3 Parameter Sweep for Cortical Magnification Transform

We tested how the shape parameters of the cortical magnification $fov, K, cover\ ratio$ affect the quality of the learned representation. Recall that: $fov$ controls the relative size of the fovea or the radius of linear sampling; $K$ controls the curviness of the periphery, the smaller (i.e. closer to $-fov$) $K$ is the degree of warping in the periphery sampling; $cover\ ratio$ approximately controls the area of sampling grid relative to the image size (see Fig.A.2). When each view is generated, we sampled uniformly in the specified range of $cover\ ratio$. We systematically varied these three parameters and examined the test and SimCLR accuracy of the learned representation (Tab. A.1). For these

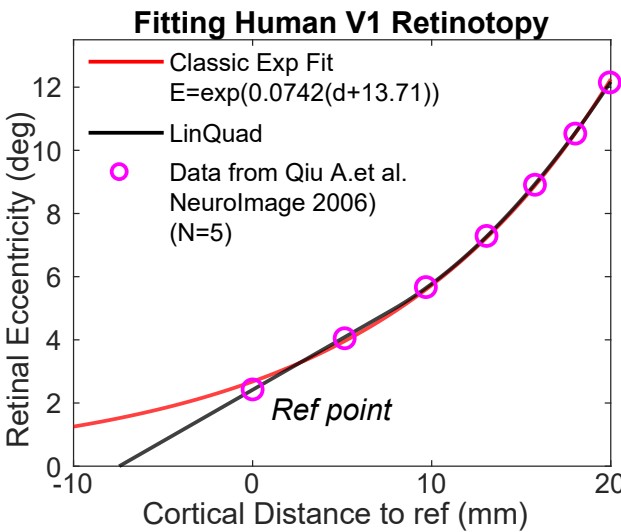

Figure A.1: **Fitting of Human V1 Average Cortical Magnification**. Data are obtained from (Qiu et al., 2006), averaged across 5 subjects and both hemifields. The x axis showed the distance to reference point (the cortical point for 2.43° radius stimuli), the y axis showed the retinal eccentricity.

experiments, we disabled the Gaussian blur and kept the color jittering, flip and random gray scale transform in place.

We found that using a larger $fov$ value generally yielded a better representation: $fov = 45$ had higher testing accuracy and higher SimCLR accuracy, compared to those of $fov = 15$. For the flatness of periphery controlled by $K$, a higher $K$ value generally resulted in a better representation. These two trends both suggest that the warped sampling doesn't necessarily improve the test accuracy and that reducing the warping by increasing the $fov$ size or increasing the $K$ will both improve the test performance. We think this is due to the domain shift introduced by the warped peripheral image: testing data were not warped. From this analysis we reasoned that if cortical magnification is the augmentation that facilitates human self-supervised learning, then the contrastive learning of visual representation might be applied to the less warped foveal and near-peripheral vision, instead of the whole visual field.

Most interestingly, comparing different sizes of sampled views ($cover\ ratio$), we found that the range $[0.05, 0.35]$ consistently yielded the highest representation quality. In this range of $cover\ ratio$, the views are local parts of the image with small overlap. In contrast, for the ranges $[0.01, 1.5]$ or $[0.05, 0.7]$, the field of views are larger, resulting in more overlaps between views, rendering the SimCLR task easier. This interpretation is in line with the consistent increase of SimCLR accuracy from $[0.01, 0.35]$, $[0.05, 0.35]$, $[0.05, 0.7]$, to $[0.01, 1.5]$ across all the $fov$ and $K$ values. Notably, it was also consistent across all $fov$ and $K$ value that the range $[0.01, 0.35]$ resulted in a lower performance than the range $[0.05, 0.35]$. We interpreted this as follows: when a local patch is too small, it will not be informative or distinguishable for the object identity. For example, a patch of white fur sampled from the image of a dog is not strongly associated with the dog category Ullman et al. (2016). As a result, encouraging similar representations between these tiny patches or between tiny patches and larger patches may harm the categorical representation of objects.

In summary, this parameter sweep suggests that the effectiveness of SimCLR training is sensitive to the scale of the sampled views.

| Cover Ratio | Testing Accuracy | | | | | | SimCLR Accuracy | | | | | |
|---|---|---|---|---|---|---|---|---|---|---|---|---|
| **fov=15, K=** | -15 | -7.5 | 5 | 20 | 35 | 50 | -15 | -7.5 | 5 | 20 | 35 | 50 |
| [0.01, 0.35] | NA | 73.4 | 76.5 | 78.1 | 78.6 | 79.2 | NA | 54.6 | 58.2 | 61.6 | 61.5 | 64.6 |
| **[0.05, 0.35]** | NA | 74.4 | 76.9 | 77.9 | 79.7 | 79.5 | NA | 62.1 | 68.7 | 73.1 | 76.4 | 75.9 |
| [0.05, 0.7] | NA | 71.4 | 74.1 | 75.7 | 76.3 | 76.4 | NA | 79.9 | 84.2 | 85.8 | 88.6 | 89.0 |
| [0.01, 1.5] | NA | 68.0 | 70.3 | 71.5 | 71.5 | 71.1 | NA | 86.5 | 89.1 | 90.3 | 90.9 | 91.1 |
| **fov=30, K=** | -15 | -7.5 | 5 | 20 | 35 | 50 | -15 | -7.5 | 5 | 20 | 35 | 50 |
| [0.01, 0.35] | | | | 79.2 | | | | | | 65.7 | | |
| **[0.05, 0.35]** | | | | 80.0 | | | | | | 75.4 | | |
| [0.05, 0.7] | | | | 76.3 | | | | | | 89.1 | | |
| [0.01, 1.5] | | | | 72.2 | | | | | | 91.4 | | |
| **fov=45, K=** | -15 | -7.5 | 5 | 20 | 35 | 50 | -15 | -7.5 | 5 | 20 | 35 | 50 |
| [0.01, 0.35] | 79.1 | | 79.0 | 79.6 | 79.0 | 79.2 | 66.0 | | 65.4 | 68.7 | 67.6 | 68.4 |
| **[0.05, 0.35]** | 79.8 | | 79.8 | 80.2 | 80.1 | 80.0 | 75.8 | | 79.0 | 80.7 | 79.6 | 81.9 |
| [0.05, 0.7] | 77.4 | | 76.7 | 77.6 | 76.6 | 77.1 | 89.0 | | 89.5 | 91.1 | 90.8 | 91.1 |
| [0.01, 1.5] | 72.5 | | 72.1 | 71.5 | 71.9 | 72.3 | 91.4 | | 91.9 | 91.8 | 91.8 | 92.2 |

Table A.1: **Parameter Tuning for the Cortical Magnification Transform**. We reported the test accuracy and SimCLR accuracy in left and right panel. The three row blocks correspond to the three $fov$ values, $15, 30, 45$. In each block, the columns correspond to the different $K$ values, and rows correspond to different range of *cover ratio*s.

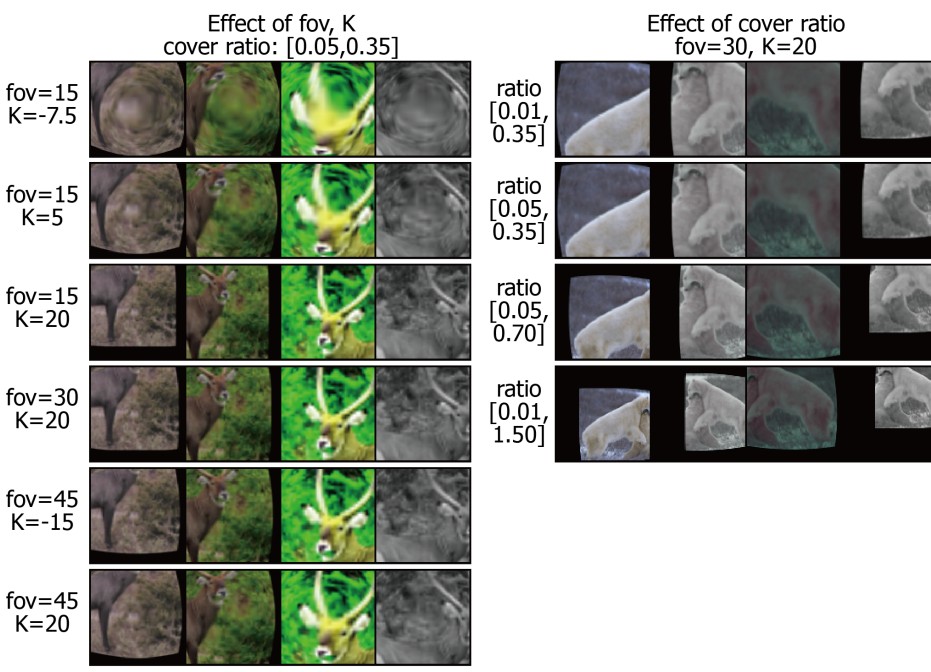

Figure A.2: **The Visual Effect of Magnification Parameters**. Views were generated using the full pipeline of Sec. A.3, Four views were sampled using the same random seed for each set of parameter $fov, K, cover\ ratio$. Left column examined the effect of shape parameters $fov$ and $K$, right column examined the effect of size parameter *cover ratio*

### A.4 Hyperparameters for SimCLR Training

For the model, we used the ResNet-18 architecture, with a 512-dimensional representation space and a 1 hidden-layer MLP as projection head. We used a 256-dimensional projection space for calculating the SimCLR loss. For the SimCLR objective, 2 views are generated per image and the soft-max temperature for the view classification is set as 0.07.

For training, Adam optimizer is used with an initial learning rate 0.0003 and a Cosine Annealing schedule (Loshchilov & Hutter, 2016), while the weight decay was set as 0.0001. All training was conducted with single GPU (TeslaV100 32G), using batch size 256. Training terminated at 100 epochs, which usually took 3-6 hrs per run.

### A.5 CO2 Emission Related to Experiments

Experiments were conducted using a private infrastructure, with a carbon efficiency around 0.432 kg $CO_2$ eq/kWh. Around 1053 hours of computation (network training) was performed on hardware of type Tesla V100-SXM2-32GB (TDP of 300W). Total emissions are estimated to be 136.47 kg $CO_2$ eq. Estimations were conducted using the Machine Learning Impact calculator presented in Lacoste et al. (2019).

### A.6 Training Process of Foveation as Blur Augmentations

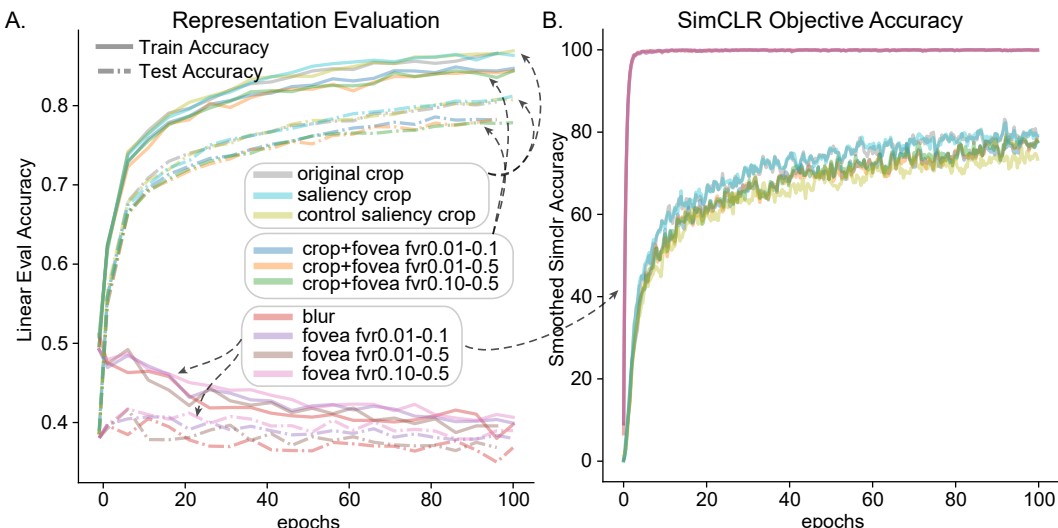

Figure A.3: **Evolution of representation quality and SimCLR objective during training of Experiment 1**. **A.** Linear probe evaluation of the representation throughout training, training and test accuracy are shown in solid and dashed lines. **B.** SimCLR Accuracy through out training. The experiments are binned into three groups: with crops, with crop+foveation, and with foveation/blur only. The SimCLR accuracy saturated rapidly for all runs without random crops, and their representation quality stopped improving or even degraded.

## A.7 Distribution of Augmented Images

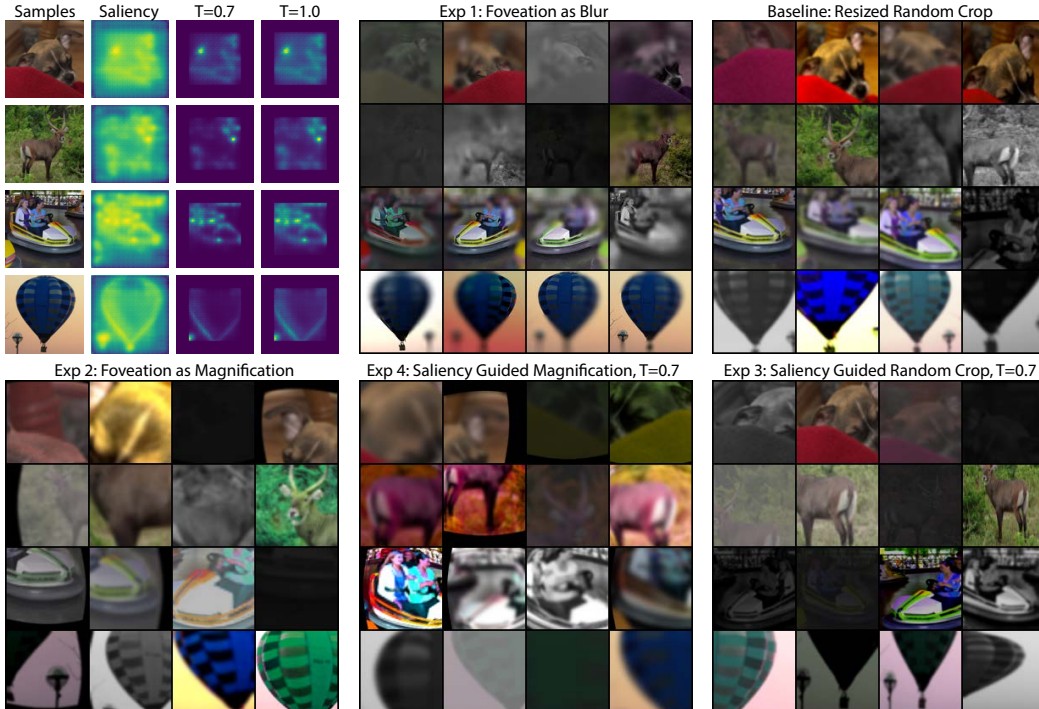

Figure A.4: **Distribution of Augmented Images in Experiments**. Four images sampled from STL-10 unlabeled set are shown as examples, the corresponding saliency map and fixation densities are shown to their side. We visualize four views for each image for each training setting: Foveation as blur, without crop; Baseline setting, resized random crops; Foveation as Magnification, without crop; saliency guided sampling for magnification; saliency guided sampling for random crops. The variation and focus of each augmentation pipeline can be seen from this comparison.

