# OpenReview forum: "On the use of Cortical Magnification and Saccades as Biological Proxies for Data Augmentation"
_NeurIPS.cc/2021/Workshop/SVRHM — SVRHM 2021 Poster_

### Official Review · Reviewer_ya1c · 2021-10-26
**Interesting paper on a potential computational benefit of eye movements and foveation for learning**

**Rating:** 7
**Confidence:** 4

**Review:**

The authors use foveation and a saccade-like sampling of images as naturalistic data augmentation techniques for self-supervised learning instead of the standard random image crops and color jittering. They find that saccade-like sampling and deforming the image to mimic cortical magnification lead to performance comparable to the standard augmentation techniques, or slightly better. Although it would have been more impactful to find truly improved performance, or more human-like characteristics using these naturalistic augmentation techniques, the fact that they can be used to rival standard methods is interesting, and proposes a new potential role for eye movements and foveation during learning that deserves further research.

Major :
-	None, the paper is well written, and the results/analysis seem solid and mostly support the claims made by the authors (but see below for a minor quibble).

Minor:
-	As far as I understand, foveation and saccades are only employed for training, but testing is done on full images? In this case, it is as though the network is trained in a biologically inspired manner, but tested without in a different regime. I understand that this is the standard in self-supervised learning of course, but it might be worth mentioning that foveation and saccades are always present in biological systems and are not just a data-augmentation tool.
-	Calling a 0.3% improvement in accuracy a “robust benefit to training” is an overstatement in my opinion. I think it is better qualified as “similar performance”.

---

### Official Review · Reviewer_qZg5 · 2021-10-31
**Modest results, but an interesting research direction**

**Rating:** 7
**Confidence:** 4

**Review:**

The paper seeks a more biologically plausible version of the image augmentations used in self-supervised learning. The guiding intuition is that the brain gets to experience "the same image" in many different variants by fixating at different locations in the scene; contrastive-like learning objectives could push the neural representations of these differently-foveated versions to all be represented as similarly as possible. This is a valuable idea, and the paper makes a sound initial test of whether the image changes caused by fixation changes (blur and magnification) can work as effective augmentation methods within a SimCLR training framework in a ResNet18 architecture. They test the effects of blur and magnification separately. They find that spatially-varying blur alone was a very poor augmentation method (networks did not train). Magnification (i.e. differential spatial sampling densities across the image) worked about exactly as well as the more standard augmentation method of taking cropped subregions from images. They further pursue "biologically plausible" augmentations by selecting foci for the magnification or cropping based on saliency maps (predictive of human fixations). Strictly sampling from only the highest-saliency regions helped the SimCLR learning objective (though I'm not sure what the interpretation of this is), but harmed the "representation quality" as measured by how well the learned features could support object classification (although some compromise can be made between these two metrics).

It doesn't appear that either cortically-inspired blur, or magnification, or behaviourally-motivated saliency sampling, substantially improves self-supervised learning. However, the paper shows that in at least some combinations (magnification and sampling), these brain-inspired implementations are just as good as more standard ML augmentation methods (spatially uniform cropping with random location sampling). So the intuition behind the work remains plausible. The rationale is well written and the methods and results are clear (given page limitations).

---

### Official Review · Reviewer_uFwN · 2021-10-31
**Interesting first exploratory study on biologically relevant mechanisms for self-supervised learning of visual representations.**

**Rating:** 8
**Confidence:** 4

**Review:**

This paper investigates two interesting hypotheses: 1) foveation-based transformations and saccade-like active sampling mechanisms may be the natural augmentations used by biological visual systems to learn in a self-supervised manner. 2) implementing such operations in a standard SSL-based setup may yield stronger performance than the standard "random crop, uniform sampling" technique of current contrastive SSL.

Pros
- Excellent fit with the topics of the SVRHM workshop
- Clear, insightful writing, study well-motivated
- Paper presents a variant of SimCLR that is more biologically plausible and yet works just as well as the traditional "engineered" setup in a fairly restricted testbed

Cons
- The implementation of foveation (two variants) and saccades is compelling; however these feel "engineered" in a similar way to existing pipelines (considering hyperparameters, etc.) I found statements like "compare [our method] against their engineered counter-part augmentations like random crops" to be a bit strong, as these methods are still engineered.
- Though completely adequate for SVRHM workshop, the dataset (STL-10) baseline architecture (ResNet 18) are limited. Even the baseline of SimCLRv2 is eclipsed by better contrastive methods. However, the field is moving quickly and it's unreasonable to expect the authors to implement all bleeding-edge SSL variants or more sophisticated architectures. I just point this out as a potential area of expansion.

Overall, I thought this was a novel and interesting workshop submission. I hope that the authors decide to release code that implements their variants of spatial-varying blur and cortical magnification operations. I like that the SimCLR accuracy ("intrinsic difficulty of the augmentation") was included as a metric in addition to train/test accuracy; there were some interesting insights that foveation as adaptive blur saturated this SimCLR accuracy metric.

I think that this paper really opens up the door to study more biologically relevant mechanisms for SSL. Nice work.

Typos
- "o.o. d" in section 4

---

### Official Review · Reviewer_rehE · 2021-11-01
**Nice motivation and some interesting preliminary experiments, but some results need to be fleshed out more and applications/impact remain unclear.**

**Rating:** 6
**Confidence:** 4

**Review:**

This paper attempts to find more biologically plausible methods of generating the "augmentations" that are utilized in nearly every recent method to train deep networks through self-supervised learning. Specifically, it has been shown in previous work that one of the most important augmentations used to generate these "positive examples" is the random crop (i.e. learning invariance by making representations for two different crops similar). However, random cropping is a somewhat engineered way to generate "views" of the same image and there is an open question as to whether there are more biologically plausible or natural mechanisms for generating the random crop augmentation.  The authors explored this idea by casting the random crop as the more perceptually relevant process of foveation combined with active saccades. They then implemented models of this process to generate natural augmentations and trained self-supervised networks using these augmentations to see how this compared to the more artificial random crop trained networks.

Positives:
- I think the motivation for this work is valuable as the image augmentation piece of self-supervised learning is definitely artificially engineered and the development of more biologically relevant augmentations definitely has the possible potential to improve and inform these models.
- Experiment 2 is a nice result showing that an implementation of foveation as cortical magnification can replace random cropping and still produce equivalently good self-supervised models (as measured by linear classification accuracy).
- Experiment 3 is a nice idea and result showing that you need at least a certain entropy for the fixation distribution to get a good classification accuracy and there is a small effect showing that there may be in fact an optimal fixation density (T=0.3-4.5) where test accuracy can be improved over the uniform sampling method normally used in random cropping.
- The paper is well written and easy to understand

Areas for Improvement:
- Experiment 1 was a negative result and while this is good to know it’s not that surprising or interesting because the power of the random crop is in the introduction of scale and content variation that you don’t get just with blurring (so it did not seem like this could replace random crop in the first place).
- The only method of evaluation used is the classification accuracy on the test set and another classification accuracy (simclr accuracy) for distinguishing the augmentations from other images. While these results are needed to show that the foveated cortical magnification can achieve similar results to the random crop, there is no further analysis about how this differs in terms of the learned representation. Specifically, other than the speculation that this may be a way for the brain to create “augmentations” if there is no real performance improvement over the random crop, what are the other reasons for including the cortical magnification in future learning methods? Does using this augmentation setup produce more robust representations or perceptually relevant representations in the trained network? Without these results, it is unclear what the impact of this work really will be.
- In Experiment 3, the [0.3%-0.7%] gain when using the correct temperature interval for sampling fixations should be validated more rigorously. Is that a statistically significant change? Or does it produce better predictions in a subset of test images? More analysis must be shown to really justify that claim that there really is a “Sweet spot” because it seems more like the test accuracy just plateaus.
- Again for experiment 3 I would also like to see more analysis of the learned representations for different temperature levels. For example, for the sweet spot model that uses temperature of 0.3 vs a uniformly sampled model, is there a high representational similarity between the models or is there something significantly different?

Overall, I think the work is of good quality and the idea of trying to find more biologically relevant “augmentations” seems interesting and this work can definitely be built on. (On this point, I would hope the authors would publish their code and experiments if the paper is published). The authors address good points for further work in the discussion, but I wish there was some more analysis of the learned representations other than linear classification accuracy because it is hard to interpret whether the cortical magnification + more saliency based sampling made a notable difference, or simply achieved a similar model (to the random crop) through a more biologically plausible method. Because of this it's hard to see the impact of this work from both a machine learning and perception point of view.

---

### Decision · Program_Chairs · 2021-11-02

Accept (Poster)